# Perceived stress across population segments characterized by differing stressor profiles—A latent class analysis

Finn Breinholt Larsen[ID][1]\*, Mathias Lasgaard[1,2], Morten Vejs Willert[3], Jes Bak Sørensen[ID][1,4]

**1** DEFACTUM, Central Denmark Region, Aarhus, Denmark, **2** Department of Psychology, University of Southern Denmark, Odense, Denmark, **3** Department of Occupational Medicine, Danish Ramazzini Centre, Aarhus University Hospital, Aarhus, Denmark, **4** AIAS, Aarhus Institute of Advanced Studies, Aarhus University, Aarhus, Denmark

\* finn.breinholt@stab.rm.dk

**Data Availability Statement:** The data contain potentially sensitive information and that there is a possibility of deductive disclosure. Therefore data may not be shared publicly according to the Danish

## Abstract

### Objective

We aimed to 1) identify distinct segments within the general population characterized by various combinations of stressors (stressor profiles) and to 2) examine the socio-demographic composition of these segments and their associations with perceived stress levels.

### Methods

Segmentation was carried out by latent class analysis of nine self-reported stressors in a representative sample of Danish adults (N = 32,417) aged 16+ years. Perceived stress level was measured by the Perceived Stress Scale (PSS).

### Results

Seven classes were identified: Class 1 was labeled Low Stressor Burden (64% of the population) and the remaining six classes, which had different stressor combinations, were labeled: 2) Burdened by Financial, Work, and Housing Stressors (10%); 3) Burdened by Disease and Death among Close Relatives (9%); 4) Burdened by Poor Social Support and Strained Relationships (8%); 5) Burdened by Own Disease (6%); 6) Complex Stressor Burden Involving Financial, Work, and Housing Stressors (2%); and 7) Complex Stressor Burden Involving Own Disease and Disease and Death among Close Relatives (2%). Being female notably increased the likelihood of belonging to Classes 2, 3, 5, and 7. Higher age increased the likelihood of belonging to Class 3. Low educational attainment increased the likelihood of belonging to Classes 5 and 6. A significant difference was observed in perceived stress levels between the seven latent classes. Average PSS varied from 9.0 in Class 1 to 24.2 in Class 7 and 25.0 in Class 6.

Data Protection Law, section 10 (https://www.datatilsynet.dk/media/7753/danish-data-protection-act.pdf). The de-identified data will be made available upon request to DEFACTUM, Public Health and Health Services Research, Central Denmark Region (hvordanhardudet@rm.dk) with an appropriate, restricted use data agreement in place.

**Funding:** The study was funded by the Velliv Foreningen, grant number 19–0176 (https://www.vellivforeningen.dk/). The grant was received by FBL. The funders had no role in the study design, data collection and analysis, decision to publish or preparation of the manuscript.

**Competing interests:** The authors have declared that no competing interests exist.

## Conclusion

Latent class analysis allowed us to identify seven population segments with various stressor combinations. Six of the segments had elevated perceived stress levels but differed in terms of socioeconomic composition and stressor combinations. These insights may inform a strategy aimed at improving mental health in the general population by targeting efforts to particular population segments, notably segments experiencing challenging life situations.

## Introduction

Stress is a major cause of psychological distress and disease, making it a significant public health issue [1]. Recent studies indicated that the majority of the general population continues to experience low levels of perceived stress [2–5]. However, a concerning rise was recorded in the proportion of individuals encountering high levels of perceived stress, particularly among younger generations [5, 6]. This trend is likely exacerbated by the negative impact of the COVID-19 pandemic on population mental health and well-being [7].

This situation warrants intensified population level efforts. An effective public health strategy to counter stress should be based on knowledge of 1) the main sources of stress in contemporary societies and 2) how sources of stress cluster and affect stress levels in segments of the population with various stressor profiles. Below, stressors are understood as any physical, psychological, or social event or condition that may cause stress in individuals and ultimately challenge physical health and mental well-being [8].

Previously, we used a variable-centered approach to explore the main sources of stress in contemporary societies. Our findings showed that among a general population own disease, work situation, and poor social support were the primary sources of perceived stress among a wide range of work-related and non-work-related stressors [9, 10]. However, the results also testified to the need for a comprehensive approach to stress since most stressors had an effect on perceived stress level [10]. In this study, we examined the second topic by adopting a person-centered approach to identify distinct groups of individuals within the general population who share similar stressor patterns. Specifically, we examined the size of each population segment, the characteristics of their stressor profiles, the socio-demographic composition of the segments, and their associations with perceived stress levels. We paid particular attention to identifying general population segments in which the combination of stressors was associated with high stress levels, highlighting a need for interventions targeting these groups.

Consistent with our previous studies, the transactional stress model (TSM), proposed by Lazarus and Folkman [11], and the stress process model (SPM), proposed by Pearlin [12], provided the theoretical backdrop of the study. In conjunction, the two theories support a perception of stress as a psychological, but socially generated and structured, phenomenon [9]. According to the TSM, stress arises when a situation is perceived as threatening or challenging and demands strain or exceeds the individual's ability to cope adequately. The SPM claims that stress results from a social process where stressors are distributed between groups and individuals on the basis of social characteristics. Both theories leave it to empirical investigation to determine which stressors are the primary sources of stress and how stressors cluster in certain groups and individuals across populations.

More recent theoretical developments have added to our understanding of the importance of the combined effects of multiple stressors on an individual's well-being. *Stressor diversity* refers to the extent to which stressor events are spread across multiple domains of life, such as

work-related stress, family conflicts, financial pressures, and health issues [13, 14]. Research has shown that experiencing greater stressor diversity is linked to better affective well-being than experiencing the same amount of stressors within a single domain [13]. The *overpowering hypothesis* predicts that older adults will respond more strongly than younger people to unpleasant events affecting multiple life domains (complex events) [15]. The *multiple burden hypothesis* predicts that combining a paid job, being married, and having children is likely to have negative effects on women's health [16]. The *role overload* model proposes that individuals who need to balance the demands and obligations of different social roles may generate "overloads" that prejudice their health [17]. The *stress proliferation* theory posits that stress in one domain of a person's life may result in stress "spillover", affecting other domains [18].

In conjunction, these theories highlight the complex nature of stress and the importance of considering multiple stressors potentially impacting an individual's well-being. They also suggest that interventions aimed at reducing stress should take into account the diverse range of stressors that people may be facing and any potential interactions between them.

Various research methods are available for person-oriented research [19]. One is *latent class analysis* (LCA), which we used in this study. LCA is a statistical method used to identify unobserved subgroups within a population [20]. LCA rests on the assumption that the observed relationships between a group of indicators can be explained by the existence of an unobserved categorical latent variable with K categories where each category represents a latent class. When the population is divided into subgroups based on the latent variable, the indicators within each subgroup become statistically independent (local independence assumption). In other words, the subgroups will be internally homogeneous with respect to the observed indicators but differ from other subgroups. The main output from a LCA is the number and size of latent classes and the response probability of each indicator given that a person belongs to a certain class. In this study, LCA allowed us to identify distinct population segments with distinct stressor profiles.

LCA is based on a stochastic model and individuals are not deterministically assigned to a latent class. Instead, an individual belongs to each of the K latent classes with a probability ranging from 0 to 1. Consider two individuals who are both more likely to belong to class k1 than to one of the other classes and that the first person has a higher probability of belonging to the class than the second person. In this case, the first person's response pattern can be considered more "typical" of that class than the second person's response pattern [21].

LCA was used to examine the clustering of occupational stressors and their association with background characteristics and work-related outcomes, including mental health and self-reported stress [22–32]. Furthermore, LCA has been used to identify patterns of stress exposures within certain population segments. For instance, it was used to examine stress patterns among older black Americans and explore how these patterns were related to depressive symptoms [33]; and among youth in residential care to investigate their association with major psychiatric problems [34]. However, to the best of our knowledge, the present study is the first to examine the clustering of stressors and their association with perceived stress in a general population.

## Methods

### Study design and data collection

We used data from the 2017 health survey "How are you?", implemented by the Central Denmark Region. "How are you?" is a representative, cross-sectional survey of the population aged 16 years and above. It is based on a stratified random sample of people living in the Central Denmark Region as per 1 January 2017. The sample was drawn from the Danish Population

Register. A total of 52,000 people were invited to participate. Participants were invited to complete a web-based or postal questionnaire [35]. Data collection started on February 3rd 2017 and ended on May 3rd 2017. Those who failed to respond to the invitation received up to four reminders. The response rate was 62% (32,417), and 80% of the respondents completed the web questionnaire. The study was approved by the Danish Data Protection Agency (r. no. 2012-58-0006) and the Central Denmark Region (r. no. 1–16–02-593-16). Danish law requires no formal ethical approval of survey and register-based studies from an ethics committee or other research oversight (§ 14 section 2) [36]. Furthermore, Danish law does not require consent from parents or guardians for the participation of minors above the age of 15 in surveys. Each participant received written information about the purpose of the study and was informed that by answering the questionnaire, they consented to participate. All parts of the survey were implemented in accordance with relevant guidelines and regulations along with the approval. Data from the survey were linked with national administrative registers using a unique personal identification number assigned to all Danish citizens [37]. Register data included age, gender, and ethnic background. To reduce sampling and non-response bias, we applied weights constructed by Statistics Denmark using a model-based calibration approach and including socio-demographic characteristics, income, social benefits, and healthcare utilization in both the descriptive and model-based analyses [35, 38].

## Variables

The questions used to assess perceived stressors cover major life events, chronic stressors, and daily hassles and were adapted from the Life Event Questionnaire [39]. The set of questions was originally selected for the Danish National Birth Cohort [40–42] and included in "How are you?", with a few modifications. The initial question was: "Within the past 12 months, have you felt burdened by some of the following things?" The questionnaire covers work situation, financial circumstances, housing conditions, relationship with partner, relationship with family and friends, own disease, disease among close relatives, and deaths among close relatives. The response categories were "no", "yes, a little", "yes, partly" or "yes, a lot". In the present study, the four response categories were dichotomized into "no/a little" and "partly/a lot". Another potential stressor, lack of perceived social support, was assessed using the following question: "Do you have someone to talk to if you have problems or need support?" The response format was "yes, always", "yes, most of the time", "yes, sometimes" or "no, never or almost never". Here, we dichotomized the response categories into "always/mostly" and "sometimes/never or almost never".

Age and sex were assessed using a combination of self-reported and register data. Ethnic background was defined using the Danish Civil Registration System [37]. Educational attainment was self-reported and categorized as low (0–10 years), medium (11–15 years), or high (15+ years) according to the Danish version of the International Standard Classification of Education [43]. Students were categorized according to their expected graduation level. Self-reported data were used to assess work status, cohabitation status, and whether or not the participants lived with children aged 0–15 years.

Level of perceived stress was assessed by the 10-item Perceived Stress Scale (PSS) [44]. Based on Lazarus' stress model [45, 46], the PSS is a global stress measure for the extent to which events or situations are perceived as demanding or threatening. The ten items ask how often in the past month life was appraised as unpredictable, uncontrollable, and overloaded (e.g., "In the last month, how often have you been upset because of something that happened unexpectedly?") [44]. The items were scored from 0 to 4 ("never", "almost never", "sometimes", "fairly often", or "very often"). An increasing sum score (range 0 to 40) indicates an

increasing perceived stress level [44]. PSS does not have validated cut-off values for determining high or low levels of perceived stress [44]. However, a number of studies have proposed cut-off values ranging from 15 to 18 for high levels of perceived stress [47–50]. PSS has satisfying psychometric properties [51, 52] and has been used in a number of population-based studies. In the present study, Cronbach's α indicated that the scale had a high internal consistency (α = 0.87). We report PSS as a continuous variable and as a dichotomized variable with a cut-off of 18.

Before conducting the analyses, data were screened for missing values. The proportion of missing values on perceived stressors varied between 6.3% (lack of perceived social support) and 9.8% (work situation). For the eight items adapted from the Life Event Questionnaire, missing values were treated as "no" if respondents had answered at least one question. Consequently, 28,201 observations (87%) had complete data for all eight items. Imputation was performed for 3462 observations (6%) with one to seven missing responses, while 2053 observations (7%) with missing values across all eight variables remained unchanged. The social support question had 2177 missing responses (7%). In total, there were 30.133 observations with complete data on the nine perceived stressor items after imputation.

Missing data for the PSS items ranged from 4.3% (item 1) to 5.1% (item 8). When fewer than four out of ten PSS items were missing, the mean of the available items was used to compute the scale score. If four or more items were missing, the PSS score was considered missing. This approach resulted in PSS scores for 29,476 observations (91%) with complete data for all ten items, and for 1549 observations (4.8%) with one to three missing items, while 1392 observations (4.3%) did not receive a PSS score.

The LCA and subsequent analyses were conducted using complete cases. The results were compared to those derived from the full sample (32,417) using full information maximum likelihood (FIML), which managed missing data directly in the likelihood function. No substantial differences were noted in cluster size or indicator prevalence, but models based on FIML showed poorer relative fit and lower entropy values compared to models based on complete cases (0.53 with FIML versus 0.59 for complete cases)

## LCA approach

The model-based analysis in this study comprised three steps: (1) segments of the general population with different stressor profiles were identified using LCA; (2) the socio-demographic composition of the segments was examined; and (3) the relationship between stressor profiles and perceived stress was analyzed. All analyses were conducted using Latent GOLD 6.0.

In the first step, we estimated LCA models that had between one and ten latent classes. The final model was selected based on statistical criteria (optimal fit) and interpretability. For model comparison, we used the Akaike Information Criterion (AIC) and the Bayesian Information Criterion (BIC). Both are measures of relative fit and seek to balance model complexity against sample size. In both measures, a lower value indicates a better model fit. The BIC prefers simpler models (fewer classes) than the AIC [53]. BIC is generally considered the more reliable of the two measures when the sample size is large as it protects against overfitting [54]. Entropy was used to describe the ability of the selected model to uniquely assign individuals to a certain class (class separation), where a high value equals good class separation. LCA rests on the assumption of local independence between indicators. Bivariate residuals (BVR) were examined as a means of describing possible sources of misfit in the selected model. A high BVR for a pair of variables indicates residual local dependency that causes model misfit. If theoretically justified, a model can be modified to account for local dependence between two variables.

In the second step, the sociodemographic composition of the stressor segments found in the first step was described. The description included the distribution of each group by age, sex, ethnic origin, educational attainment, work status, cohabitation status, and children in the home. A multinomial logistic regression model was used to predict how the individual variable affected the probability of belonging to the different stressor groups when adjusting for the other covariates.

In the third step, we examined the variation in perceived stress across stressor segments using one-way analysis of variance (ANOVA). For the second and third steps, we used procedures developed in recent years to relate latent class membership to external variables that adjust for the fact that each subject is assigned to a latent class with less than 100% certainty, thereby reducing bias in the estimates due to classification uncertainty [55–58]. For the bivariate analysis of the sociodemographic composition of the stressor segments and the multivariate multinomial logistic regression model, we applied the two-step correction method outlined by Bakk and Kuha [58]. In the first step, the selected LCA model was estimated without incorporating external variables. In the second step, the parameters of the LCA model were fixed at their estimated values and the relationship between the covariates and the latent classes was assessed. The two-step method is particularly effective for minimizing bias from classification uncertainty in models that includes covariates, especially when the LCA measurement model exhibits poor separation between latent classes (low entropy).

For the ANOVA model, we employed the three-step method with the correction procedure initially introduced by Bolck et al. [56] and later refined by Bakk et al. [57]. In the first step, the chosen latent class model was estimated without external variables. In the second step, each respondent was assigned to their most likely latent class. The third step involved estimating the relationship between the latent classes and the distal outcome (PSS), while adjusting for bias due to classification uncertainty.

## Results

### Sample description

Table 1 describes the distribution of the study sample. The mean age was 47.6 years (standard deviation (SD) 19.3 years, range 16–102 years), with an equal distribution of men and women. High educational attainment was reported by 29% of the participants, 59% were employed, and 66% were married or cohabitating, while 28% were residing with at least one child aged 0–15 years. Nine out of ten sampled people were of Danish ethnic origin. The most commonly reported stressors were related to work, own disease, and financial circumstances, whereas stressors related to relationships with family or friends were less prevalent.

### Selection of latent class model

Table 2 summarizes the fit criteria for latent class models with one to ten classes. After consulting the fit indices and prior to selecting the optimal class model, we considered whether the classes were distinct and substantively meaningful. According to the BIC criterion, the seven-class model had the best relative fit, whereas the AIC did not reach a minimum level. The seven-class model was chosen. This choice was underpinned by the fact that the model can be given a meaningful interpretation based, among others, on experience from clinical practice in occupational medicine.

The selected model has some shortcomings. While a classification error of 0.18, a dissimilarity index value of 0.027, and a minimum class size of 2% are considered acceptable, the model does not fully explain all interrelationships between variables, particularly between poor social support and own disease, as indicated by a bivariate residual of 4.02 (max(BVR)).

**Table 1. Characteristics of study population.**

| Characteristics | N | Weighted prevalence (%) |
|---|---|---|
| Sex | | |
| Male | 15,105 | 50 |
| Female | 17,312 | 50 |
| Age (mean (SD)) | | 47.6 (19.3) |
| Age (years) | | |
| 16–24 | 3,790 | 15 |
| 25–34 | 3,423 | 15 |
| 35–44 | 4,619 | 15 |
| 45–54 | 5,914 | 17 |
| 55–64 | 5,845 | 15 |
| 65–74 | 5,702 | 14 |
| 75+ | 3,124 | 9 |
| Educational attainment | | |
| Low (0–10 years) | 4,580 | 16 |
| Medium (11–15 years) | 17,050 | 54 |
| High (15- years) | 8,518 | 29 |
| Cohabitation status | | |
| Married/cohabitating | 22,314 | 66 |
| Single | 8,396 | 34 |
| Ethnic origin | | |
| Danish | 30,469 | 89 |
| Other Western | 847 | 4 |
| Non-Western | 1,101 | 6 |
| Work status | | |
| Working | 17,775 | 59 |
| Non-working | 12,218 | 41 |
| Living with child(ren) aged 0–15 years | | |
| No | 20,145 | 72 |
| Yes | 7,452 | 28 |
| Burdened by the following stressors within the last 12 months (partly/a lot) | | |
| Financial circumstances | 4,020 | 16 |
| Housing conditions | 2,117 | 9 |
| Work situation | 4,748 | 17 |
| Relationship with partner | 2,134 | 8 |
| Relationship with family and friends | 1,587 | 6 |
| Disease | 5,223 | 17 |
| Disease among close relatives | 4,277 | 14 |
| Deaths among close relatives | 2,445 | 8 |
| Lack of perceived social support | 4,159 | 15 |
| Number of reported perceived stressors (mean (SD)) | | 1.10 (1.45) |

SD = standard deviation

Residual values above 3.84 are statistically significant at the 0.05 level. Although we could potentially have improved the fit of the model to the data by relaxing the assumption of conditional independence between these two variables, we opted not to do so due to a lack of theoretical justification. The second largest residual in the seven-class model was 3.08.

**Table 2. Fit statistics and diagnostic criteria for latent class analyses.**

| Number of latent classes | Number of parameters estimated | LL | BIC | AIC | max(BVR) | Entropy | Class Error | DI | Smallest class size (%) |
|---|---|---|---|---|---|---|---|---|---|
| 1 | 9 | -97,490 | 195,072 | 194,997 | 4873.86 | 1.00 | 0.00 | 0.260 | 100 |
| 2 | 19 | -90,751 | 181,698 | 181,540 | 1315.62 | 0.64 | 0.07 | 0.099 | 22 |
| 3 | 29 | -90,005 | 180,308 | 180,067 | 253.13 | 0.58 | 0.12 | 0.068 | 12 |
| 4 | 39 | -89,624 | 179,650 | 179,327 | 68.70 | 0.55 | 0.16 | 0.057 | 6 |
| 5 | 49 | -89,382 | 179,269 | 178,862 | 34.80 | 0.54 | 0.17 | 0.040 | 4 |
| 6 | 59 | -89,267 | 179,141 | 178,652 | 11.25 | 0.60 | 0.16 | 0.032 | 3 |
| 7 | 69 | -89,167 | **179,044** | 178,472 | 4.02 | 0.59 | 0.18 | 0.027 | 2 |
| 8 | 79 | -89,132 | 179,077 | 178,422 | 2.03 | 0.57 | 0.19 | 0.025 | 1 |
| 9 | 89 | -89,108 | 179,134 | 178,395 | 1.85 | 0.57 | 0.19 | 0.024 | 1 |
| 10 | 99 | -89,086 | 179,193 | 178,371 | 1.05 | 0.53 | 0.22 | 0.022 | 0.3 |
| 11–15 | Not well identified | | | | | | | | |

LL = log-likelihood; BIC = Bayesian information criterion; AIC = Akaike information criterion; max(BVR) = the largest bivariate residual in the pairwise cross-table of the nine indicators; DI = Dissimilarity index

In addition, the ability of the model to allocate individuals to a specific class was moderate, as evidenced by its 0.59 entropy value. While values above 0.8 are considered satisfactory [59], this value suggests that the model has several borderline cases or observations that are difficult to categorize into a single class. Despite these shortcomings, we believe that the model yields valuable insights.

## Characteristics of latent classes

Class proportions and the estimated probabilities of having experienced any particular stressor within the past 12 months are shown in Table 3. Class 1 was characterized by low probabilities across all stressors, whereas the other six classes reported high or moderate probabilities in one or more domains. All seven classes were labelled according to the stressor profiles (Table 3). Of note, Class 4 had a less distinctive profile than the other classes with the highest class-specific probability of 0.46 (poor social support). In contrast, Classes 2–3 and 5–7 all had "signature variables" with probabilities ranging from 0.76 to 0.94, while class-specific probabilities in Class 1 were low, ranging from 0.01 to 0.09.

## Population characteristics of latent stressor classes

Table 4 shows the socio-demographic composition of the seven latent classes and the results of the multivariate, multinomial logistic regression analysis.

In the regression model, several factors were found to influence the probability of belonging to the classes:

*Sex*: Females were more likely to belong to Classes 2, 3, 5, and 7 and less likely to belong to Class 6, compared with Class 1.

*Age*: Higher age increased the likelihood of belonging to Class 3 but decreased the likelihood of belonging to Classes 2, 6, and 7, compared with Class 1

*Educational attainment*: Low educational attainment increased the likelihood of belonging to Classes 5 and 6, and decreased the likelihood of belonging to Class 3, compared with Class 1.

*Marital status*: Being married or cohabitating increased the likelihood of belonging to Class 3 and decreased the probability of belonging to Classes 2, 5, 7, and particularly Class 4, compared with Class 1

**Table 3. Latent class results characterizing seven stressor profiles (class proportions and class-specific probabilities of stressors).**

| | Latent Class | | | | | | |
|---|---|---|---|---|---|---|---|
| **Class** | 1 | 2 | 3 | 4 | 5 | 6 | 7 |
| **Assigned label** | Low Stressor Burden | Burdened by Financial, Work, and Housing Stressors | Burdened by Disease and Death Among Close Relatives | Burdened by Lack of Social Support and Strained Relationships | Burdened by Own Disease | Complex Stressor Burden Involving Financial, Work, and Housing Stressors | Complex Stressor Burden Involving Own Disease and Disease and Death Among Close Relatives |
| Class proportion | 0.64 | 0.10 | 0.09 | 0.08 | 0.06 | 0.02 | 0.02 |
| Item-response probabilities | | | | | | | |
| Financial circumstances | 0.04 | **0.80** | 0.05 | 0.17 | 0.11 | ***0.92*** | **0.59** |
| Housing conditions | 0.02 | 0.42 | 0.03 | 0.12 | 0.02 | ***0.77*** | 0.41 |
| Work situation | 0.07 | **0.57** | 0.11 | 0.28 | 0.27 | ***0.77*** | 0.47 |
| Relationship with partner | 0.02 | 0.12 | 0.06 | 0.38 | 0.00 | ***0.56*** | 0.36 |
| Relationship with family and friends | 0.01 | 0.05 | 0.04 | 0.29 | 0.04 | ***0.65*** | 0.47 |
| Disease | 0.04 | 0.30 | 0.19 | 0.27 | ***0.94*** | **0.54** | **0.76** |
| Disease among close relatives | 0.02 | 0.15 | **0.77** | 0.21 | 0.12 | 0.29 | ***0.92*** |
| Deaths among close relatives | 0.03 | 0.07 | 0.34 | 0.10 | 0.07 | 0.10 | ***0.60*** |
| Perceived social support | 0.09 | 0.19 | 0.09 | 0.46 | 0.22 | ***0.56*** | 0.45 |
| Number of reported perceived stressors (mean) | 0.32 | 2.69 | 1.67 | 2.28 | 1.78 | 5.16 | 5.04 |

Item-response probabilities > 0.5 in bold to facilitate interpretation. With each item, the class with the highest response probability is in italic.

*Ethnic origin*: Non-Danish ethnic origin increased the likelihood of belonging to Classes 2, 4, and particularly Classes 6 and 7, compared with Class 1.

*Employment status*: Having a non-working status increased the probability of belonging to Classes 2, 5, 7, and particularly Class 6, while decreasing the probability of belonging to Classes 3 and 4, compared with Class 1.

*Living with children*: Living with children aged 0–15 years increased the likelihood of belonging to Classes 2, 4, 6, and 7 and decreased the probability of belonging to Classes 3 and 5, compared with Class 1.

## The association between latent classes and perceived stress level

The variation in perceived stress levels across latent classes is reported in Fig 1. A significant difference was observed in stress levels between the seven latent classes. For PSS; the average varied from 9.0 in Class 1 to 25.0 in Class 6. Correspondingly, the proportion with a high stress level varied from 7% in Class 1 to 91% in Class 6.

To assess the importance of the increased perceived stress level in Classes 2 to 7 compared with Class 1, we calculated Cohen's d and utilized his rule of thumb, as expanded by Sawilowsky [60], for interpretation. The effect size of membership of Class 3 compared with Class 1

**Table 4. Demographics and multinomial logistic regression results for covariates by latent class of stressors (intercept omitted from table).**

| | Latent Class | | | | | | | | | | | | | |
|---|---|---|---|---|---|---|---|---|---|---|---|---|---|---|
| Class | 1 | | 2 | | 3 | | 4 | | 5 | | 6 | | 7 | | |
| Assigned label | Low Stressor Burden | | Burdened by Financial, Work, and Housing Stressors | | Burdened by Disease and Death Among Close Relatives | | Burdened by Lack of Social Support and Strained Relation-ships | | Burdened by Own Disease | | Complex Stressor Burden Involving Financial, Work, and Housing Stressors | | Complex Stressor Burden Involving Own Disease and Disease and Death Among Close Relatives | | Wald test p-value |
| Characteristics | % | OR | % | OR | % | OR | % | OR | % | OR | % | OR | % | OR | |
| Mean age (SD), years | | 49,4 | | 34,9 | | 57,6 | | 40,3 | | 58,9 | | 36,3 | | 46,0 | |
| **Age, years** | | | | | | | | | | | | | | | |
| 16–24 | 14 | | 26 | | 6 | | 21 | | 0 | | 20 | | 11 | | |
| 25–34 | 13 | | 30 | | 6 | | 14 | | 4 | | 34 | | 16 | | |
| 35–44 | 14 | | 21 | | 10 | | 25 | | 11 | | 20 | | 13 | | |
| 45–54 | 17 | | 16 | | 17 | | 23 | | 18 | | 17 | | 24 | | |
| 55–64 | 16 | | 6 | | 20 | | 11 | | 29 | | 8 | | 23 | | |
| 65–74 | 17 | | 1 | | 23 | | 5 | | 16 | | 1 | | 8 | | |
| 75+ | 10 | | 0 | | 18 | | 1 | | 21 | | 0 | | 4 | | |
| Age (per 5-year increase) | | 1.0 | | **0.8** | | **1.1** | | **0,9** | | **1.0** | | **0.8** | | **0.9** | p<0.001 |
| **Sex** | | | | | | | | | | | | | | | |
| Male | 53 | 1,0 | 47 | 1,0 | 37 | 1,0 | 49 | 1,0 | 36 | 1,0 | 50 | 1,0 | 37 | 1,0 | p<0.001 |
| Female | 47 | 1,0 | 53 | **1,3** | 63 | **1,9** | 51 | 1,0 | 64 | **1,7** | 50 | 0,9 | 63 | **1,4** | |
| **Ethnic origin** | | | | | | | | | | | | | | | |
| Danish | 93 | 1,0 | 82 | 1,0 | 95 | 1,0 | 83 | 1,0 | 95 | 1,0 | 75 | 1,0 | 70 | 1,0 | p<0.001 |
| Other Western | 4 | 1,0 | 8 | **2,3** | 3 | 1,0 | 6 | **2,3** | 1 | **0,7** | 6 | **3,4** | 8 | **3,0** | |
| Non-Western | 4 | 1,0 | 10 | **1,8** | 2 | **0,7** | 11 | **2,4** | 4 | 0,9 | 19 | **4,2** | 23 | **6,3** | |
| **Educational attainment** | | | | | | | | | | | | | | | |
| High (15+ years) | 15 | 1,0 | 12 | 1,0 | 16 | 1,0 | 14 | 1,0 | 29 | 1,0 | 25 | 1,0 | 26 | 1,0 | p<0.001 |
| Medium (11–15 years) | 55 | 1,0 | 56 | 1,1 | 54 | **0,9** | 53 | 0,9 | 51 | 1,3 | 54 | 1,7 | 47 | 0,9 | |
| Low (0–10 years) | 30 | 1,0 | 32 | **0,9** | 30 | **0,8** | 33 | 1,1 | 20 | **1,6** | 21 | **1,9** | 28 | **1,5** | |
| **Work status** | | | | | | | | | | | | | | | |
| Working | 63 | 1,0 | 61 | 1,0 | 53 | 1,0 | 75 | 1,0 | 27 | 1,0 | 31 | 1,0 | 40 | 1,0 | p<0.001 |
| Non-working | 37 | 1,0 | 39 | **2,1** | 47 | **0,9** | 25 | **0,9** | 73 | **2,9** | 69 | **5,7** | 60 | **2,5** | |
| **Living with spouse/cohabitant** | | | | | | | | | | | | | | | |
| Single | 31 | 1,0 | 48 | 1,0 | 26 | 1,0 | 48 | 1,0 | 38 | 1,0 | 56 | 1,0 | 41 | 1,0 | p<0.001 |
| Married/cohabitating | 69 | 1,0 | 52 | **0,8** | 74 | **1,3** | 52 | **0,4** | 62 | **0,8** | 44 | **0,6** | 59 | **0,6** | |
| **Living with child(ren) aged 0–15 years** | | | | | | | | | | | | | | | |
| No | 73 | 1,0 | 61 | 1,0 | 81 | 1,0 | 56 | 1,0 | 89 | 1,0 | 64 | 1,0 | 74 | 1,0 | p<0.001 |
| Yes | 27 | 1,0 | 39 | **1,3** | 19 | **0,9** | 44 | **2,1** | 11 | **0,8** | 36 | **2,0** | 26 | **1,4** | |

\* = p<0,05. OR = odds ratio. Each odds ratio is adjusted for the remaining variables in the model.

was slightly below medium (d = 0.41). Membership of Class 2 demonstrated a large effect size (d = 1.11), whereas membership of Class 4 and Class 5 showed very large effect sizes (d = 1.48; d = 1.44), and membership of Class 6 and Class 7 exhibited exceptionally large effect sizes (d = 2.35; d = 2.23).

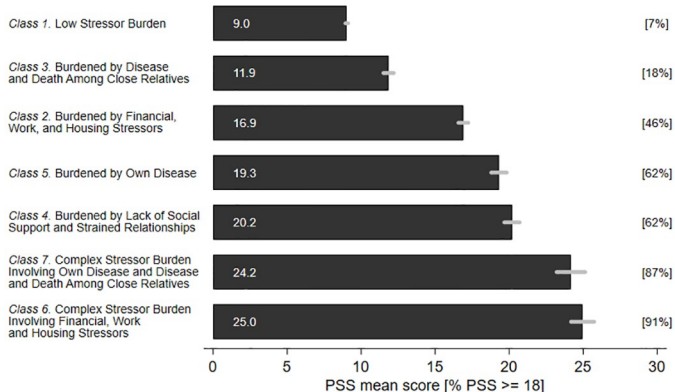

**Fig 1. Perceived stress levels across latent classes.**

## Discussion

This study used LCA to identify population segments with different stressor profiles. Seven latent classes were identified. These seven classes offer a detailed understanding of how various common sources of stress clustered across the general Danish population. Moreover, further analyses revealed marked differences in socio-economic composition and perceived stress levels across these classes. Altogether, this study presents novel insights that may be used to effectively address mental health concerns in the general population, focusing on groups in the population who find themselves in particularly challenging situations. In the following sections, we discuss the distinctive characteristics of each segment, ordered from lowest to highest perceived stress level. Furthermore, we apply a life-course perspective to the identified segments to enhance our understanding of the dynamic interaction between the segments.

### Stressor profiles in the population

*Class 1* comprised almost two-thirds of the population. In this segment, the stressor burden was low in all domains, and the segment had the lowest level of perceived stress of all seven classes. Compared with the entire adult population, the segment had a moderate over-representation of men, persons of Danish ethnic background, persons in work, and persons who were married/cohabiting.

*Class 3* comprised 9% of the population. In this segment, the stressor burden was very high for disease among close relatives and high for deaths among close relatives but low in the other domains. The perceived stress level was somewhat elevated compared with Class 1. The group was characterized by a predominance of women and older people. This group also had the largest proportion of married/cohabiting individuals. The group included people who cared for an ill partner, parent, child, friend, etc., or who mourned the loss of a loved one.

It is estimated that around one in six people of the adult population in the 28 EU member states are informal caregivers [61], which has been identified as a chronic stressor that places them at risk for physical and emotional problems [62, 63]. On this backdrop, it is noteworthy that only one in five citizens in Class 3 had a high level of perceived stress. Thus, the majority of Class 3 appeared to cope well with their caregiver burden or the worries and grief associated with illness and death of a loved one. Moreover, they had managed to balance this with other commitments, perhaps because they did not experience other major stressors in their lives. It is, however, important not to trivialize the caregiver burden. In combination with other

stressors, it can pose a serious threat to mental well-being. Older people may generally be better than younger people at navigating around stressors in their lives [64–66]. However, many older people have fewer resources than younger people to withstand adversity and may therefore be more easily "knocked over" by unforeseen stressors [67]. Older individuals who are burdened by caregiving are therefore less robust if their circumstances change for the worse, e.g., if they are affected by illness. Moreover, grief in adults is associated with depression and other psychological disorders [68, 69].

Points of attention: An increasing number of people require or will require long-term care. Society is heavily dependent on informal caregivers and should recognize and support their efforts. Half of Class 3 had a job, and for this group, it is important to focus on improving work-life balance. It is important to acknowledge the grief of those who have lost a loved one and provide psychological treatment for those experiencing complicated grief reactions.

*Class 2* included 10% of the population. The stressor burden was very high for finances, work, and housing. The perceived stress level was significantly elevated compared with Class 1. Class 2 was the youngest of the seven classes as eight out of ten people were 16–44 years old. Women, people residing with dependent children, and people who were single were moderately overrepresented, whereas people of a non-Danish background appeared twice as often as in the general population. Class 2 included young people who were in education or at the beginning of their professional careers and who were struggling with the challenges of doing well in the educational system, establishing themselves on the labor market, finding a partner, becoming parents, and obtaining affordable and satisfactory housing. Most young people experience a high degree of well-being, but the transition into adulthood can be challenging for some [70].

Researchers have pointed out that the transition into adulthood has become much more protracted in recent decades, and that it is now often delayed until the late twenties or early thirties [71]. Additionally, the more or less standardized passages into adulthood of earlier times have been replaced by a more diversified and highly individualized 'in-between' period, referred to as emerging adulthood by some researchers [72, 73]. Finding employment remains an important condition upon which other transitions into adulthood build [71], but changes in the labor market have increased the number of insecure and 'precarious' jobs, which disproportionately affect young people [74]. Several longitudinal studies indicated that entry into a labor market with precarious employment is associated with an increased risk of subsequent mental health problems [75–77].

Points of attention: A group of predominantly young people are struggling to create a stable financial basis for their existence. It is important to maintain a political focus on conditions that can ensure young people a safe transition into adulthood, including access to education, a labor market with secure jobs, and affordable housing.

*Class 5* comprised 6% of the population. Almost everyone in this class reported being burdened by disease. The perceived stress level was much greater than in Class 1. Class 5 corresponded in composition to Class 3 with a predominance of women, older, and married/cohabiting individuals. However, the group had a lower level of education and a higher percentage of non-working people (three out of four did not work).

Most diseases may cause a degree of uncertainty, disruption of life routines, and lack of control, at least for a period of time [78, 79]. Chronic disease, in particular, requires changes across psychological, social, and physical aspects of a person's life [80]. Therefore, living with one or more chronic conditions is often perceived as stressful. This may be due to functional limitations and symptoms related to the disease itself (disease burden). However, a growing body of research also indicates that treatment of the disease often places a burden on the patient as it includes attending appointments, undergoing medical investigations and

procedures, self-monitoring, adopting lifestyle advice, navigating healthcare systems, taking medications, and enduring any adverse effects [81–83]. In addition, chronic disease frequently results in financial burdens (although this is less common in Class 5 than in Class 7 where disease coincides with multiple other stressors), presents existential challenges, and entails fundamental changes in self-perception (e.g. described as 'biographical disruption') [84].

Points of attention: An increasing number of people are diagnosed and treated for chronic conditions [85] and chronic disease is a major source of stress [9, 10]. It is important that healthcare remains aware of patients' psychological well-being. Access to mental support is required for patients who need it. It is also important to minimize the treatment burden, address patients' priorities, support self-management, and underpin healthy behaviors. Furthermore, it is essential to implement policy measures aimed at enhancing employment opportunities for individuals with health challenges. Employers must also acknowledge that illness can be a significant source of stress among their employees.

*Class 4* included 8% of the population with a stressor burden mainly characterized by lack of social support and strained relationships. The perceived stress level was very high compared with Class 1. Class 4 was the second youngest of the seven classes. It spanned people from young adulthood to middle-aged people, with an equal distribution of men and women. Persons of non-Danish background appeared almost twice as often as in Class 1. Class 4 had the highest prevalence of people living with a dependent child.

As mentioned above, Class 4 had a less marked profile than the other classes with the highest class-specific probability of 0.46 (poor social support). Therefore, caution is needed when interpreting these findings. Even so, Class 4 seems to include people burdened by social relations–relationship with partner, relationship with family and friends, and poor social support. Forming and maintaining close relationships–with parents, siblings, friends, partners, children, etc.–is an important part of life for most people [86] and a source of joy and satisfaction but also a source of social negativity (including frustration, conflict, pain, and suffering) and grief. A review of stress research in couples found that conflicts and tensions often arose between the partners from expressed differences in goals, attitudes, needs, desires, habits, and personality traits of one's partner. Conflicts also arose due to insufficient support for household tasks, stress coping, or compatibility issues among partners [87].

Transitions in close relationships are a well-known source of stress [88]. For instance, the transition into parenthood is among the most challenging experiences for committed partners [89, 90]. The changing of close relationships from harmonious to conflict-ridden may result in the loss of a crucial source of social support, impacting mental well-being negatively, while the end of close relationships may lead to separation distress. A recent research overview of the psychology of close relationships emphasized the importance of studying relationships within their contexts (e.g., relationship type, life stage, and culture) [86].

Points of attention: At any point in time, a certain part of the population lives in strained relationships with their partner, parents, or another primary person; or suffers the consequences of social disconnectedness (e.g. lack of social support). Supportive environments can help people through periods of relational difficulties, and the promotion of social inclusivity, trust, and interconnectedness within various settings, such as residential areas, educational institutions, and workplaces, is important for enhancing well-being and social health. For some individuals, seeking professional assistance, such as counseling or psychotherapy, has proved highly beneficial. Ensuring equal and convenient access to these services for those in need is paramount for fostering a supportive and healthy society.

*Class 7* comprised 2% of the population. This was the first of two classes with a complex stressor burden. This class was characterized by a particularly high burden of own disease, disease among and death of close relatives, and financial burdens. The perceived stress level was

exceptionally high compared with Class 1. Class 7 corresponded in composition to Class 5 (Burdened by Own Disease) and partly to Class 3 (Burdened by Disease and Death among Close Relatives) with a majority of women, married/cohabiting persons, and a high proportion with low education. The age composition, however, was significantly younger with an age profile dominated by middle-aged and young people. Furthermore, Class 7 had a significantly higher proportion of people of non-Danish ethnic origin and people with children in the household. This indicates that Class 7 was not just an amalgamation of Class 3 and 5—"double-hit people"—but a group that has followed a different developmental path.

Points of attention: This class consists of people in a very vulnerable situation with an increased risk of developing–or who have already developed–pathological stress reactions. Prevention and treatment strategies have recently been suggested for people who experience multiple stressors in regular, daily life [91]. These strategies include keeping close relationships active, public and media educational programs aimed at enhancing individual resilience by utilization of social networks, and, if treatment is needed, psychotherapy in the form of interpersonal psychotherapy and cognitive behavioral therapy.

Based on the stressor profile, a public health strategy aimed at reducing the stressor burden in Class 7 should prioritize initiatives within the healthcare system. These efforts should address social support gaps and mental health challenges related to disease management and treatment, as well as the unique stressors associated with being a relative or caregiver. Such initiatives must also take into account the high proportion of individuals in this group who have relatively limited resources and face complex stressors, including financial burdens.

Evidence from a meta-analysis [92] highlights the short-term benefits of mental health interventions for informal caregivers, such as preventive psychology, recovery-focused therapy, and psychoeducation. However, the authors also stress the need to weigh potential adverse effects against these benefits, noting the absence of a gold standard for improving mental health outcomes in informal caregivers.

Given this context, we believe it is crucial to provide targeted support for both patients and their relatives in managing the mental health challenges that arise during and after interactions with the healthcare system. Such support could enhance their overall resilience and reduce the cumulative stressor burden experienced by this vulnerable group.

*Class 6* included 2% of the population. This was the second of two classes with a complex stressor burden. This group was characterized by a particularly high burden due to financial circumstances, housing conditions, work situation, strained relations with their partner, family, and friends, but also of own disease and poor social support. The perceived stress level was exceptionally high compared with Class 1.

Class 6 was dominated by young people with a particularly large overrepresentation of individuals aged 25–34 years. Individuals in this class faced personal adversity on many fronts and seemed to have embarked on a path of numerous hardships early in life. According to a recent research overview, the social context of young people's mental health has far-reaching consequences [93]. Findings link peer and parenting problems in youth to adult mental health problems [94]. A cohort study [95] showed that young people who face social adversities, such as parental illness, unemployment, and housing shortages, tend to develop mental health problems. Studies of young adults emphasized the importance of supportive relationships with family and friends [96]. While stable, supportive peer relationships provided protection, family and peer problems raised stress levels [97, 98].

Points of attention: A small group of predominantly young people is on a risky and difficult developmental path and needs help to cope with the challenges in their lives. Similar to Class 7, this group requires a dedicated and comprehensive preventive and therapeutic intervention from society.

Based on the stressor profile, a public health strategy aimed at reducing the stressor burden in Class 6 should adopt a dual approach. One branch should focus on children and young people in vulnerable situations, while the other should target predominantly younger adults on the margins of the labor market and in precarious circumstances.

For children and young people, initiatives should emphasize family-centered interventions and be integrated within day care services, schools, and community-based social programs. These efforts could provide a foundational support system for addressing stressors early in life. For younger adults, initiatives should combine social support with labor market interventions, aiming to reduce the compounded stressors associated with unemployment and economic instability.

Recent evidence from an umbrella review of community-based interventions [99] highlights the strongest outcomes for initiatives addressing financial insecurity and providing welfare support, both of which were shown to improve mental health outcomes. Incorporating such evidence-based strategies into targeted programs could enhance their efficacy in alleviating stressor burdens for these vulnerable groups.

## A life-course perspective on stressor profiles

By adopting a life-course perspective, the results of the present study may be interpreted dynamically, cf. Fig 2. From infancy to old age, individuals adapt to various challenges and stressors at different life stages [100]. Developmental research on mental health points to the heterogeneity in the developmental processes of the life course [101]. The specific stressor patterns identified in this study likely reflect both this uniformity and variation. Most of the time, most people are able to stay within their mental comfort zone as they navigate and negotiate their way through life (Class 1), whereas others, influenced by personal choices and external factors, will be pushed to their limits or beyond for shorter or longer periods, some even permanently (Class 2–7). While Class 1 included all age groups, Classes 2 and 6 were dominated by young people, Class 4 was dominated by young and middle-aged people, Class 7 by middle-aged people, and Classes 3 and 5 were dominated by older people. It may be hypothesized that individuals transition into and out of the segments as they move along their life course with Class 1 as a kind of default state. This hypothesis is supported by the fact that we previously found that within-person changes in stressor burden were considerable over a four-year period [10]. Furthermore, the hypothesis is in line with the theoretical framework of our study since

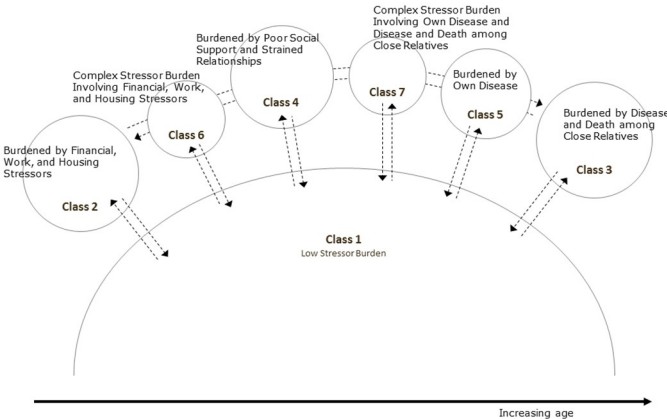

**Fig 2. A life-course perspective on stressor profiles.**

the ability to cope with stressors will most likely change during the life course and is thus related to social characteristics and life circumstances at any given time. In this perspective, the life-course approach underlines the importance of not focusing exclusively on single stress factors, such as occupational stress, in stress mitigation policies and interventions. Instead, it is crucial to adopt a comprehensive approach consistent with the previously mentioned theories; one that underscores the combined effects of multiple stressors and recognizes the importance of life-course context. It is critical to ensure that individuals do not remain stuck in a high-stress segment, causing chronic stress, or to prevent them from transitioning into another high-stress segment. This dynamic interpretation of stressor profiles should be tested in future longitudinal research.

To our knowledge, this is the first study to analyze how sources of stress cluster and associate with perceived stress in a general population using LCA. A strength of our study is that it included a large sample size (32,417 individuals) and a wide age range from a representative sample of the general population.

However, our study also carries some limitations. The included stressors are self-reported, which implies potential response bias. The analyses were also limited by the number of stressors included. Furthermore, the LCA model adopted does not fully explain all relationships between stressors. Finally, the model has low class separation, which means that it poorly predicts individuals' class affiliation and therefore has many borderline cases. To enhance the robustness the presented model, replication studies using additional data sets would be beneficial. Specifically, incorporating a broader range of perceived stressors and more diverse variables related to social support–such as material, emotional and practical support across various context like work and family–would be particularly valuable.

Theoretically, greater stressor diversity is associated with better affective well-being compared to an equivalent number of stressors concentrated within a single domain, as proposed by Koffer et al. [13]. However, our findings reveal deviations from this theoretical framework. Specifically, some classes exhibit low perceived stress levels despite low stressor diversity (e.g., Class 3), while others report high perceived stress levels despite high stressor diversity (e.g., Class 4).

Class 3, characterized by disease and deaths among relatives as the predominant stressors, reports lower perceived stress levels compared to all other classes except Class 1. In contrast, Class 4, which exhibits moderate probabilities across multiple stress domains—including higher probabilities for poor social support—reports elevated perceived stress levels.

These findings suggest that certain stressors may disproportionately influence perceived stress levels, overriding the potential benefits of diversity. Prior research [9] indicates that stressors such as personal illness, perceived social support, and work situation are the most significant contributors to perceived stress. In comparison, stressors such as disease and deaths among relatives account for a smaller proportion of the variance in perceived stress levels.

Our results underscore the importance of refining the stressor diversity theory by accounting for the differential impact of individual stressors on perceived stress. This nuance could enhance the explanatory power of the theory and provide a more comprehensive understanding of the relationship between stressor diversity and well-being.

## Conclusion

Utilizing latent class analysis, we identified seven distinct population segments characterized by different stressor profiles. Six of the segments exhibited moderately to highly elevated perceived stress levels but differed in terms of socioeconomic composition and stressor profiles. Of note, 4% of the population grappled with a complex interplay of stressors and an

exceptionally high level of perceived stress. The segments identified may be hypothesized to present a snapshot of a dynamic pattern in which individuals transition in and out of segments with differing stressor characteristics as they move along their life course. These findings may inform strategies aimed at enhancing overall mental health in the general population by addressing issues and difficulties that are unique to specific population segments facing particularly challenging life situations.

## Supporting information

**S1 File. Data availability.**
(DOCX)

## Author Contributions

**Conceptualization:** Finn Breinholt Larsen, Jes Bak Sørensen.

**Data curation:** Finn Breinholt Larsen.

**Formal analysis:** Finn Breinholt Larsen.

**Funding acquisition:** Finn Breinholt Larsen, Jes Bak Sørensen.

**Investigation:** Finn Breinholt Larsen.

**Methodology:** Finn Breinholt Larsen.

**Project administration:** Finn Breinholt Larsen.

**Resources:** Finn Breinholt Larsen.

**Writing – original draft:** Finn Breinholt Larsen, Jes Bak Sørensen.

**Writing – review & editing:** Mathias Lasgaard, Morten Vejs Willert, Jes Bak Sørensen.

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
