## [Decision Letter · Decision Letter 0]

6 Oct 2024

PONE-D-23-43197Perceived stress across population segments characterized by differing stressor profiles – a latent class analysisPLOS ONE

Dear Dr. Larsen,

Thank you for submitting your manuscript to PLOS ONE. After careful consideration, we feel that it has merit but does not fully meet PLOS ONE’s publication criteria as it currently stands. Therefore, we invite you to submit a revised version of the manuscript that addresses the points raised during the review process.

After careful consideration of the reviewers' comments and your manuscript, I believe your work has potential for publication, but requires major revisions before it can be accepted. Both reviewers have raised important points that need to be addressed to strengthen the scientific rigor and clarity of your paper.

Key areas for revision include:

1. Model Selection and Justification

2.Data Handling and Missingness

3. Results Reporting and Interpretation

4. Discussion and Implications

5. Methodology Clarification

Please provide a point-by-point response to all reviewer comments along with your revised manuscript. Ensure that all changes are clearly marked in the revised version.

I believe that addressing these concerns will significantly strengthen your paper and increase its potential impact in the field. I look forward to receiving your revised manuscript.

We look forward to receiving your revised manuscript.

Kind regards,

Piotr Janusz Mamcarz, Doctor

Academic Editor

PLOS ONE

2. We note that you have indicated that there are restrictions to data sharing for this study. For studies involving human research participant data or other sensitive data, we encourage authors to share de-identified or anonymized data. However, when data cannot be publicly shared for ethical reasons, we allow authors to make their data sets available upon request. For information on unacceptable data access restrictions, please see http://journals.plos.org/plosone/s/data-availability#loc-unacceptable-data-access-restrictions. Before we proceed with your manuscript, please address the following prompts: a) If there are ethical or legal restrictions on sharing a de-identified data set, please explain them in detail (e.g., data contain potentially identifying or sensitive patient information, data are owned by a third-party organization, etc.) and who has imposed them (e.g., a Research Ethics Committee or Institutional Review Board, etc.). Please also provide contact information for a data access committee, ethics committee, or other institutional body to which data requests may be sent. b) If there are no restrictions, please upload the minimal anonymized data set necessary to replicate your study findings to a stable, public repository and provide us with the relevant URLs, DOIs, or accession numbers. Please see http://www.bmj.com/content/340/bmj.c181.long for guidelines on how to de-identify and prepare clinical data for publication. For a list of recommended repositories, please see https://journals.plos.org/plosone/s/recommended-repositories. You also have the option of uploading the data as Supporting Information files, but we would recommend depositing data directly to a data repository if possible. Please update your Data Availability statement in the submission form accordingly.

3. In the online submission form, you indicated that [The data contain potentially sensitive information and that there is a possibility of deductive disclosure. Therefore data may not be shared publicly according to the Danish Data Protection Law, section 10 (https://www.datatilsynet.dk/media/7753/danish-data-protection-act.pdf). The de-identified data will be made available upon request to DEFACTUM, Public Health and Health Services Research, Central Denmark Region (hvordanhardudet@rm.dk) with an appropriate, restricted use data agreement in place.]. All PLOS journals now require all data underlying the findings described in their manuscript to be freely available to other researchers, either 1. In a public repository, 2. Within the manuscript itself, or 3. Uploaded as supplementary information. This policy applies to all data except where public deposition would breach compliance with the protocol approved by your research ethics board. If your data cannot be made publicly available for ethical or legal reasons (e.g., public availability would compromise patient privacy), please explain your reasons on resubmission and your exemption request will be escalated for approval.

Additional Editor Comments (if provided):

Dear Authors,

Thank you for submitting your manuscript "Perceived stress across population segments characterized by differing stressor profiles – a latent class analysis" to PLOS ONE. I have now received feedback from two expert reviewers, and I am writing to communicate my decision. After careful consideration of the reviewers' comments and your manuscript, I believe your work has potential for publication, but requires major revisions before it can be accepted. Both reviewers have raised important points that need to be addressed to strengthen the scientific rigor and clarity of your paper.

Key areas for revision include:

1. Model Selection and Justification

2. Data Handling and Missingness

3. Results Reporting and Interpretation

4. Discussion and Implications

5. Methodology Clarification

Please provide a point-by-point response to all reviewer comments along with your revised manuscript. Ensure that all changes are clearly marked in the revised version.

I believe that addressing these concerns will significantly strengthen your paper and increase its potential impact in the field. I look forward to receiving your revised manuscript.

Reviewers' comments:

Reviewer's Responses to Questions

**Comments to the Author**

1. Is the manuscript technically sound, and do the data support the conclusions?

Reviewer #1: Partly

Reviewer #2: Yes

2. Has the statistical analysis been performed appropriately and rigorously? 

Reviewer #1: Yes

Reviewer #2: Yes

3. Have the authors made all data underlying the findings in their manuscript fully available?

Reviewer #1: Yes

Reviewer #2: No

4. Is the manuscript presented in an intelligible fashion and written in standard English?

Reviewer #1: Yes

Reviewer #2: Yes

5. Review Comments to the Author

Reviewer #1: The authors use LCA to describe stressor profiles, their social patterning, and associations with perceived stress levels, among a representative sample of Danish adults. Please find comments in the attached.

Reviewer #2: 1. Model Selection Justification: While you have justified your model choice based on AIC and BIC, the entropy value (0.59) indicates some level of classification uncertainty. I suggest exploring whether relaxing the assumption of local independence could improve the fit or, alternatively, providing stronger theoretical justification for the final model selection despite the noted limitations.

2. Discussion of Stressor Profiles: The discussion of the seven classes is insightful, but it would benefit from a more detailed exploration of potential interventions or practical applications for the identified segments. For instance, how can public health strategies be tailored to address the needs of Classes 6 and 7, which exhibit the highest perceived stress?

3. Data Availability: While it is understandable that the data are subject to legal restrictions, I recommend providing additional information about the process and conditions under which data access can be granted to ensure transparency and facilitate future research.

4. Addressing Model Limitations: Consider providing more information on how future research could address the model’s limitations, especially concerning the low entropy and significant bivariate residuals. This would strengthen the paper’s robustness and guide readers on how to improve upon the current study.

5. Clarity and Conciseness: The manuscript would benefit from some editing to reduce redundancy, particularly in the results and discussion sections. Streamlining these sections would enhance readability and ensure that the key findings are presented more clearly.

6. PLOS authors have the option to publish the peer review history of their article (what does this mean?). If published, this will include your full peer review and any attached files.

Reviewer #1: No

Reviewer #2: No

---

## [Author Response · Author response to Decision Letter 0]

20 Nov 2024

Perceived stress across population segments - point-by-point response to reviewers

We would like to thank the reviewers for their thorough and valuable comments and contributions to our manuscript.

General remark: 

During the revision of the article, we discovered a data error: seven observations were mistakenly coded as “no” on the eight Life Event Questionnaire items but should have been marked as missing. After correcting this error, the number of complete cases for the nine perceived stressors decreased from 30,140 to 30,133 observations. We conducted all analysis again using the corrected data set, which did not result in significant changes in the outcomes. The tables have been updated accordingly.

Reviewer #1: 

The authors use LCA to describe stressor profiles, their social patterning, and associations with perceived stress levels, among a representative sample of Danish adults. There are important aspects to be addressed; please find some comments below:

1. For the ANOVA and multinomial regression models, authors describe using “procedures developed in recent years to relate latent class membership to external variables”. This is quite vague; please elaborate on the approach used to assign individuals to their best fit class.

Thank you for pointing this out. We have added the following text in the Methods section page 10-11:

For the bivariate analysis of the sociodemographic composition of the stressor segments and the multivariate multinomial logistic regression model, we applied the two-step correction method outlined by Bakk and Kuha [Bakk & Kuha 2018]. In the first step, the selected LCA model was estimated without incorporating external variables. In the second step, the parameters of the LCA model were fixed at their estimated values and the relationship between the covariates and the latent classes was assessed. The two-step method is particularly effective for minimizing bias from classification uncertainty in models that includes covariates, especially when the LCA measurement model exhibits poor separation between latent classes (low entropy).

For the ANOVA model, we employed the three-step method with the correction procedure initially introduced by Bolck et al. [Bolck et al. 2004] and later refined by Bakk et al. [Bakk et al. 2014]. In the first step, the chosen latent class model was estimated without external variables. In the second step, each respondent was assigned to their most likely latent class. The third step involved estimating the relationship between the latent classes and the distal outcome (PSS), while adjusting for bias due to classification uncertainty.

2. Missingness is not discussed. Did the 32,417 survey respondents have complete data on all variables of interest? Are the LCA and subsequent analyses based on this full sample of 32,417 respondents? What proportion of respondents was missing and/or excluded from analyses and what measures, if any, were taken to address this? Please discuss.

This is a very good point. We have added the following text in the Methods section page 8-9:

Before conducting the analyses, data were screened for missing values. The proportion of missing values on perceived stressors varied between 6.3% (lack of perceived social support) and 9.8% (work situation). For the eight items adapted from the Life Event Questionnaire, missing values were treated as “no” if respondents had answered at least one question. Consequently, 28,201 observations (87%) had complete data for all eight items. Imputation was performed for 3462 observations (6%) with one to seven missing responses, while 2053 observations (7%) with missing values across all eight variables remained unchanged. The social support question had 2177 missing responses (7%). In total, there were 30.133 observations with complete data on the nine perceived stressor items after imputation.

Missing data for the PSS items ranged from 4.3% (item 1) to 5.1% (item 8). When fewer than four out of ten PSS items were missing, the mean of the available items was used to compute the scale score. If four or more items were missing, the PSS score was considered missing. This approach resulted in PSS scores for 29,476 observations (91%) with complete data for all ten items, and for 1549 observations (4.8%) with one to three missing items, while 1392 observations (4.3%) did not receive a PSS score.

The LCA and subsequent analyses were conducted using complete cases. The results were compared to those derived from the full sample (32,417) using full information maximum likelihood (FIML), which managed missing data directly in the likelihood function. No substantial differences were noted in cluster size or indicator prevalence, but models based on FIML showed poorer relative fit and lower entropy values compared to models based on complete cases (0.53 with FIML versus 0.59 for complete cases).”

3. The final selected model has low-class separation and is cause for concern. While substantive interpretation and clinical relevance help justify selection of this 7-class model, additional corroboration using other model parameters would add further support. Each parameter carries its own strengths and limitations to justify its use. Suggest that authors review other information criteria (e.g. consistent Akaike Information Criterion [CAIC], adjusted Bayesian Information Criterion [aBIC]) and/or model indices (e.g., the likelihood-ratio G2 statistic) and present this information to lend further support for model selection. 

Thank you for raising this concern.

In addition to Bayesian Information Criterion (BIC) and Akaike Information Criterion (AIC) we also calculated the Akaike Information Criterion 3 (AIC3), the Consistent Akaike Information Criterion (CAIC), and the sample size adjusted BIC (SABIC). 

Like BIC, both CAIC and SABIC reached a minimum with a seven-class model, while AIC3, like AIC, did not reach a minimum. To maintain clarity, we opted not to report these additional measures of relative fit, as they did not provide significant new insights. 

4. Can the authors present results on the mean number of reported perceived stressors for each class? This data is omitted from Table 3 and could add value to the description and discussion of stressor profiles.

Thank you for bringing this to our attention. The figures were mistakenly omitted from Table 3 but have now been included.

5. Results seemingly diverge from theories linking “greater stressor diversity” to better “affective well-being”. Take Class 3, for example, where disease (and deaths) among relatives is the only major stressor and perceived stress levels and effects sizes are lower than the other 6 stressor profiles. Alternatively, consider Class 4, where probabilities are moderate (~0.3) across several stress domains (i.e., work, family, friend and partner relationships, and own disease) but quite high (~0.5) for poor social support, and perceived stress levels and effects sizes are very high. Perhaps better perceived social support is a mitigating factor. Do authors have thoughts on other factors that may promote resilience among this segment (i.e., Class 3) of the Danish population? Please comment.

Thank you for drawing attention to this interesting phenomenon. We have added this to the Discussion page 26-27.

Theoretically, greater stressor diversity is associated with better affective well-being compared to an equivalent number of stressors concentrated within a single domain, as proposed by Koffer et al. (2016). However, our findings reveal deviations from this theoretical framework. Specifically, some classes exhibit low perceived stress levels despite low stressor diversity (e.g., Class 3), while others report high perceived stress levels despite high stressor diversity (e.g., Class 4).

Class 3, characterized by disease and deaths among relatives as the predominant stressors, reports lower perceived stress levels compared to all other classes except Class 1. In contrast, Class 4, which exhibits moderate probabilities across multiple stress domains—including higher probabilities for poor social support—reports elevated perceived stress levels.

These findings suggest that certain stressors may disproportionately influence perceived stress levels, overriding the potential benefits of diversity. Prior research (Sørensen et al., 2021) indicates that stressors such as personal illness, perceived social support, and work situation are the most significant contributors to perceived stress. In comparison, stressors such as disease and deaths among relatives account for a smaller proportion of the variance in perceived stress levels.

Our results underscore the importance of refining the stressor diversity theory by accounting for the differential impact of individual stressors on perceived stress. This nuance could enhance the explanatory power of the theory and provide a more comprehensive understanding of the relationship between stressor diversity and well-being.

Please review and correct inconsistencies in reporting:

Thank you for highlighting this. We have corrected the inconsistencies. With regard to d) Table 4 has been corrected – the single and married/cohabiting categories had been switched around.

a. Abstract: “Average PSS varied from 9.0 in Class 1 to 24.2 in Class 7 and 26.0 in Class 6.” Figure 1 describes an average PSS of 25.0 for Class 6.

b. Results: “... while 29% were residing with at least one child aged 0-15 years.” Table 1 reports 28% of sample residing with at least one child aged 0-15 years. 

c. Results: “... minimal class size of 2% are considered acceptable.” Table 2 reports minimal class size of 3% for 7-class solution. 

d. Results: “Marital status: Being married or cohabitating increased the likelihood of belonging to Class 3 and decreased the probability of belonging to Classes 2, 5, 7, and particularly Class 4, compared with Class 1”. Table 4 suggests that these results reflect single status and not being married or cohabitating.

Minor grammatical errors:

e. Omitted word in bold and underlined: “Furthermore, we apply a life-course perspective to the identified segments to enhance our understanding of the dynamic interaction between the segments.”

f. Omitted word in bold and underlined: “Class 4 included 8% of the population with a stressor burden mainly characterized by lack of social support and strained relationships”.

Reviewer #2: 

1. Model Selection Justification: While you have justified your model choice based on AIC and BIC, the entropy value (0.59) indicates some level of classification uncertainty. I suggest exploring whether relaxing the assumption of local independence could improve the fit or, alternatively, providing stronger theoretical justification for the final model selection despite the noted limitations.

Thanks for this suggestion. To improve the model, we estimated a series of models in which the conditional independence assumption was relaxed for the indicators with the largest bivariate residuals. Allowing correlations between Perceived social support and Disease, as well as between Perceived social support and Relationship with family and friends, eliminated the two largest bivariate residuals and improved the model’s relative fit. However, this adjustment reduced classification accuracy, with entropy declining from 0.59 to 0.49.

Based on these findings, and since we lack a theoretical justification for relaxing the assumption of independence among these three variables, we chose to retain the original model, which we believe offers a meaningful representation of the data pattern. Instead, we note in the discussion that the limitations of the model should be considered when interpreting the results (page 25).

2. Discussion of Stressor Profiles: The discussion of the seven classes is insightful, but it would benefit from a more detailed exploration of potential interventions or practical applications for the identified segments. For instance, how can public health strategies be tailored to address the needs of Classes 6 and 7, which exhibit the highest perceived stress?

We agree and believe it to be an excellent idea for making a stronger connection to public health strategies. We have decided to focus on classes 6 and 7 as suggested and have expanded the “points-of-attention” sections for both classes (page 23 and 24).

Class 7:

Points of attention: This class consists of people in a very vulnerable situation with an increased risk of developing – or who have already developed – pathological stress reactions. Prevention and treatment strategies have recently been suggested for people who experience multiple stressors in regular, daily life (92). These strategies include keeping close relationships active, public and media educational programs aimed at enhancing individual resilience by utilization of social networks, and, if treatment is needed, psychotherapy in the form of interpersonal psychotherapy and cognitive behavioral therapy. Based on the stressor profile, a public health strategy aimed at reducing the stressor burden in Class 7 should prioritize initiatives within the healthcare system. These efforts should address social support gaps and mental health challenges related to disease management and treatment, as well as the unique stressors associated with being a relative or caregiver. Such initiatives must also take into account the high proportion of individuals in this group who have relatively limited resources and face complex stressors, including financial burdens.

Evidence from a meta-analysis (Cherak et al., 2021) highlights the short-term benefits of mental health interventions for informal caregivers, such as preventive psychology, recovery-focused therapy, and psychoeducation. However, the authors also stress the need to weigh potential adverse effects against these benefits, noting the absence of a gold standard for improving mental health outcomes in informal caregivers.

Given this context, we believe it is crucial to provide targeted support for both patients and their relatives in managing the mental health challenges that arise during and after interactions with the healthcare system. Such support could enhance their overall resilience and reduce the cumulative stressor burden experienced by this vulnerable group.

Class 6:

Points of attention: A small group of predominantly young people is on a risky and difficult developmental path and needs help to cope with the challenges in their lives. Similar to Class 7, this group requires a dedicated and comprehensive preventive and therapeutic intervention from society. Based on the stressor profile, a public health strategy aimed at reducing the stressor burden in Class 6 should adopt a dual approach. One branch should focus on children and young people in vulnerable situations, while the other should target predominantly younger adults on the margins of the labor market and in precarious circumstances.

For children and young people, initiatives should emphasize family-centered interventions and be integrated within day care services, schools, and community-based social programs. These efforts could provide a foundational support system for addressing stressors early in life. For younger adults, initiatives should combine social support with labor market interventions, aiming to reduce the compounded stressors associated with unemployment and economic instability.

Recent evidence from an umbrella review of community-based interventions (Williams & Kirkbride, 2024) highlights the strongest outcomes for initiatives addressing financial insecurity and providing welfare support, both of which were shown to improve mental health outcomes. Incorporating such evidence-based strategies into targeted programs could enhance their efficacy in alleviating stressor burdens for these vulnerable groups.

3. Data Availability: While it is understandable that the data are subject to legal restrictions, I recommend providing additional information about the process and conditions under which data access can be granted to ensure transparency and facilitate future research.

Thank you for raising this concern. Our data are fully available for researchers, who can apply for the dataset through our website. Unfortunately, the website is 

---

## [Decision Letter · Decision Letter 1]

16 Dec 2024

Perceived stress across population segments characterized by differing stressor profiles – a latent class analysis

PONE-D-23-43197R1

Dear Dr. Larsen,

We’re pleased to inform you that your manuscript has been judged scientifically suitable for publication and will be formally accepted for publication once it meets all outstanding technical requirements.

Kind regards,

Piotr Janusz Mamcarz, Doctor

Academic Editor

PLOS ONE

Additional Editor Comments (optional):

Dear Authors

After careful consideration of your revised manuscript and the changes you have made in response to the reviewers' comments, both the reviewers and I agree that your paper makes a significant contribution to the field of stress research and population health.

Kind regards

Reviewers' comments:

Reviewer's Responses to Questions

**Comments to the Author**

1. If the authors have adequately addressed your comments raised in a previous round of review and you feel that this manuscript is now acceptable for publication, you may indicate that here to bypass the “Comments to the Author” section, enter your conflict of interest statement in the “Confidential to Editor” section, and submit your "Accept" recommendation.

Reviewer #1: All comments have been addressed

Reviewer #2: All comments have been addressed

2. Is the manuscript technically sound, and do the data support the conclusions?

Reviewer #1: Yes

Reviewer #2: Yes

3. Has the statistical analysis been performed appropriately and rigorously? 

Reviewer #1: Yes

Reviewer #2: Yes

4. Have the authors made all data underlying the findings in their manuscript fully available?

Reviewer #1: Yes

Reviewer #2: No

5. Is the manuscript presented in an intelligible fashion and written in standard English?

Reviewer #1: Yes

Reviewer #2: Yes

6. Review Comments to the Author

Reviewer #1: (No Response)

Reviewer #2: (No Response)

7. PLOS authors have the option to publish the peer review history of their article (what does this mean?). If published, this will include your full peer review and any attached files.

Reviewer #1: **Yes: **Leslie S. Craig

Reviewer #2: **Yes: **Prof Dr Saad Alatrany

---

## [Editor Report · Acceptance letter]

6 Jan 2025

PONE-D-23-43197R1 

PLOS ONE

Dear Dr. Larsen, 

I'm pleased to inform you that your manuscript has been deemed suitable for publication in PLOS ONE. Congratulations! Your manuscript is now being handed over to our production team.

Kind regards, 

on behalf of

Dr. Piotr Janusz Mamcarz 

Academic Editor

PLOS ONE